# Glyoxal's impact on dry ammonium salts: fast and reversible surface aerosol browning

David O. De Haan,*[1] Lelia N. Hawkins,[2] Kevin Jansen,[3] Hannah G. Welsh,[2] Raunak Pednekar,[2,5] Alexia de Loera,[1] Natalie G. Jimenez,[1] Margaret A. Tolbert,[3] Mathieu Cazaunau,[4] Aline Gratien,[4] Antonin Bergé,[4] Edouard Pangui,[4] Paola Formenti,[4] Jean-François Doussin[4]

[1] Department of Chemistry and Biochemistry, University of San Diego, 5998 Alcala Park, San Diego CA 92110 USA
[2] Department of Chemistry, Harvey Mudd College, 301 Platt Blvd, Claremont CA 91711 USA
[3] Department of Chemistry / Cooperative Institute for Research in Environmental Sciences, University of Colorado, Boulder CO 80309 USA
[4] Laboratoire Interuniversitaire des Systèmes Atmosphériques (LISA), UMR7583, CNRS, Université Paris-Est-Créteil (UPEC) et Université de Paris, Institut Pierre Simon Laplace (IPSL), Créteil, France
[5] Deceased

Correspondence to: David O. De Haan (ddehaan@sandiego.edu)

**Abstract.** Alpha-dicarbonyl compounds are believed to form brown carbon in the atmosphere via reactions with ammonium sulfate (AS) in cloud droplets and aqueous aerosol particles. In this work, brown carbon formation in AS and other aerosol particles was quantified as a function of relative humidity (RH) during exposure to gas-phase glyoxal (GX) in chamber experiments. Under dry conditions (RH < 5%), solid AS, AS/glycine, and methylammonium sulfate aerosol particles brown within minutes upon exposure to GX, while sodium sulfate particles do not. When GX concentrations decline, browning goes away, demonstrating that this dry browning process is reversible. Declines in aerosol albedo are found to be a function of $[GX]^2$, and are consistent between AS and AS/glycine aerosol. Dry methylammonium sulfate aerosol browns 4× more than dry AS aerosol, but deliquesced AS aerosol browns much less than dry AS aerosol. Optical measurements at 405, 450, and 530 nm provide an estimated Ångstrom absorbance coefficient of -16 ±4. This coefficient and the empirical relationship between GX and albedo are used to estimate an upper limit to global radiative forcing by brown carbon formed by 70 ppt GX reacting with AS (+7.6 ×10$^{-5}$ W/m$^2$). This quantity is < 1% of the total radiative forcing by secondary brown carbon, but occurs almost entirely in the ultraviolet range.

## 1 Introduction

Brown carbon is the name given to light-absorbing organic molecules present in atmospheric aerosol. Estimates of the global direct radiative effect of brown carbon aerosol range from +0.05 to 0.27 W/m$^2$ (Tuccella et al., 2020;Laskin et al., 2015;Zhang et al., 2019;Wang et al., 2018). This absorption occurs mainly at ultraviolet (UV) and near-UV wavelengths, suppressing photochemistry in areas with high loadings (Mok et al., 2016). Limiting emissions of brown carbon aerosol and its precursor

species could provide immediate climate benefits. Approximately 30% of brown carbon is secondary (Mukai and Ambe, 1986;Hecobian et al., 2010), formed from gas-phase species often through reactions taking place in clouds, fog, and aqueous aerosol particles (Hecobian et al., 2010). Reactions between small, multi-function aldehydes such as glyoxal and ammonium
salts (Shapiro et al., 2009;Kampf et al., 2012), and oxidation reactions of phenolic species (Chang and Thompson, 2010) are two examples of aqueous-phase brown carbon formation processes.

Glyoxal uptake to deliquesced ammonium sulfate particles is rapid (Kroll et al., 2005), but is difficult to detect on dry aerosol (Corrigan et al., 2008). Glyoxal reacts to form brown carbon imidazole derivatives in solutions containing ammonium ions
(Shapiro et al., 2009;Noziere et al., 2009;Galloway et al., 2009;Yu et al., 2011;Kampf et al., 2012;Maxut et al., 2015) or primary amine species such as glycine or methylamine (De Haan et al., 2009a;De Haan et al., 2009b). While in bulk aqueous solution these reactions take hours to days (Shapiro et al., 2009;Noziere et al., 2009;Powelson et al., 2014), they can occur in minutes in aqueous aerosol particles, likely due to surface reactivity of glyoxal in its monohydrate form (De Haan et al., 2009a).

In this work, we report rapid and reversible browning of dry ammonium sulfate (AS), AS/glycine, and methylammonium sulfate (MeAS) aerosol particles upon exposure to gas phase glyoxal. This browning process is not accompanied by appreciable particle growth, and is reversed upon addition of water vapor.

## 2 Methods

### 2.1 Large chamber experiments

CESAM is a 4.2 $m^3$ temperature- and pressure-controlled, stirred, stainless steel chamber (Wang et al., 2011) with solar simulator lamps (Harris et al., 2017), held just above ambient pressure with automated flows of high purity $O_2$ and liquid $N_2$ boil-off at a respective 20/80 v/v ratio. The chamber gas phase contents were monitored by a relative humidity (RH) sensor (Vaisala HMP234 Humicap), long-path FTIR spectroscopy (Bruker Tensor 37, 182.5 ±0.5 m path length (Wang et al., 2011), glyoxal integrated 2950-2700 $cm^{-1}$ band intensity = 6.34 $\times10^{-18}$ cm $molec^{-1}$ (Eurochamp, 2010)), and high-resolution proton
transfer reaction mass spectrometry (PTR-MS, KORE Tech. Series II, inlet temperature 100°C, proton transfer reactor P = 1.64 mbar, glow discharge P = 1.94 mbar, PTR entry voltage = 400 V, E/N ratio = 130). Polydisperse seed particles (TSI 3076 atomizer) were diffusion dried before addition to the dry chamber, then continuously sampled through a 1 m Nafion drying tube to scanning mobility particle sizing (SMPS, TSI, 20 – 900 nm) and cavity-attenuated phase shift single-scattering albedo (450 nm CAPS-ssa, Aerodyne) (Onasch et al., 2015) spectrometers. A particle-into-liquid sampler (PILS, Brechtel
Manufacturing) sampled $N_2$-diluted chamber aerosol through an activated carbon denuder, into a capillary waveguide UV/vis spectrometer (LWCC-100, 0.94m pathlength). Water vapor was added in bursts from a stainless steel boiler (Wang et al., 2011), and chamber RH was subsequently stabilized by routing inlet $N_2$ flow through a heated high-purity water bubbler. A

droplet spectrometer (Palas Welas Digital 2000, 0.5 to 15 µm diam., on chamber flange) (Wang et al., 2011) extended the size range of detected aerosol into supermicron particles. CAPS-ssa aerosol extinction and scattering signals were zeroed against

filtered chamber air every 5 min to ensure that any gas-phase species absorbing light at 450 nm does not influence measurements, and averaged to SMPS scan frequency. SMPS number and concentrations and PTR-MS signals were corrected for dilution caused by flows into the chamber. SMPS size distributions were also corrected using size-dependent wall losses measured for AS particles in the chamber.

**2.2 Small chamber experiments**

Additional experiments were conducted in a 300 L collapsible Tedlar chamber. Aerosols were generated from 0.1% w/w aqueous solutions (TSI 3076 atomizer) and diffusion dried (except in experiments on "wet" aerosol). Glyoxal production was monitored at the inlet by absorbance at 405 nm using a cavity ringdown (CRD) spectrometer and a cross section of $4.491 \times 10^{-20}$ $cm^2$ molecule$^{-1}$ (Volkamer et al., 2005b). Glyoxal concentrations at the chamber outlet were measured in test experiments

to determine wall loss rates ($\sim 6.7 \times 10^{-4}$ s$^{-1}$). Glyoxal inlet concentrations, flow mixing ratios, and wall loss rates were then used to estimate glyoxal chamber concentrations. Aerosol particles were sampled via diffusion driers by Q-AMS (Aerodyne), CAPS-ssa (Aerodyne, 450 nm), SMPS (TSI), CRD (405 and 530 nm) (Ugelow et al., 2017), and photoacoustic spectrometers (PAS, 405 and 530 nm) (Ugelow et al., 2017), all of which were periodically baselined through filters to eliminate interferences by gas-phase species. RH sensors monitored humidity levels at the aerosol inlet, chamber outlet, and dried chamber outlet

flows. Water vapor was added in certain experiments by passing inlet flows through Nafion humidifiers.

**2.3 Chemicals**

Reagents were used as received from Sigma-Aldrich unless otherwise mentioned. Solutions for aerosol generation were generated by dilution of glycine (>99%) to 5 mM, AS (>99%) to 1.2 – 10 mM, or sodium sulfate (≥99%) to 7 mM in deionized

water (>18 MΩ, ELGA Maxima). MeAS was generated by mixing methylamine and sulfuric acid (Mallinckrodt) solutions at a 2:1 molar ratio; after dilution to 6.3 mM, solution pH was 4.5. Gas-phase glyoxal was generated by heating solid mixtures of glyoxal trimer dihydrate (Fluka, >95%) and $P_2O_5$ (99%) to 110-150 °C; the glyoxal produced was flushed into the chamber with dry $N_2$ (Volkamer et al., 2009).

**3. Results**

Chamber experiments where aerosol particles were exposed to gas-phase glyoxal are summarized in Table 1.

**3.1 Dry AS and AS/glycine aerosol (experiments 1-4)**

Experiment 1, where dry AS aerosol was sequentially exposed to 0.05 ppm and then 0.50 ppm glyoxal at $t = 4:47$ and $5:16$ h, respectively, is summarized in Figure 1. Both glyoxal additions were detectable by PTR-MS at $m/z$ 31 and 59. The $m/z$ 59

signal, however, is elevated in the clean and dry chamber before glyoxal is added, indicating background interference by another chemical species or its fragment in the mass spectrometer. SMPS data, which has been corrected for wall losses and for dilution, shows no observable aerosol growth after either glyoxal gas addition. This lack of observed growth at <5% RH is consistent with previous studies under very dry conditions (Kroll et al., 2005;De Haan et al., 2017). However, as optical and chemical measurements described below will show, this lack of growth does not indicate a lack of glyoxal reactivity.

The addition of 0.05 ppm glyoxal gas at $t$ = 4:47 h triggered a short-lived drop in albedo by 0.034 that was observed by CAPS-ssa at 450 nm. A second, larger glyoxal addition (0.50 ppm) at $t$ = 5:16 h caused albedo to plummet to 0.75, a change 7× greater than the first. Even though particle sizes did not increase after either glyoxal addition, the significant albedo declines indicate that glyoxal reactions rapidly produced light-absorbing products at dry AS particle surfaces. Over the next 30 minutes, as glyoxal gas-phase concentrations decreased by half (likely to due to chamber wall losses), aerosol albedo recovered proportionately, indicating that this surface brown carbon formation under dry conditions is fully reversible.

At $t$ = 5:46 h (Figure 1), the chamber was humidified to 50% RH, a level which would not deliquesce the AS seeds (Biskos et al., 2006) but which may produce as many as 1-2 monolayers of adsorbed water at aerosol surfaces consisting of solid AS (Denjean et al., 2014;Romakkaniemi et al., 2001) or glyoxal reaction products. (Hawkins et al., 2014). Humidification to 50% RH caused significant changes to both the gas and aerosol phases. Glyoxal PTR-MS signals and aerosol albedo returned within a few minutes back to near-baseline levels, while the dried aerosol mass measured by SMPS jumped downward by 15%. The loss of gas-phase glyoxal, again without aerosol growth, suggests that water greatly accelerated glyoxal loss rates to the steel chamber walls. The simultaneous albedo recovery and SMPS mass loss indicate that humidification destroyed all brown carbon products that absorb 450 nm light, converting some fraction of them to gas-phase products. A proposed mechanism for this process is discussed below. It is significant that no browning was observed in PILS-sampled aerosol at any point during experiment 1, presumably due to the same mechanism occurring during wet sampling.

The 15% aerosol mass loss upon humidification to 50% RH is surprising, given that no corresponding mass gain was recorded during exposure to glyoxal under dry conditions. However, the lack of mass gain under dry conditions cannot be interpreted as a lack of glyoxal uptake or reactivity, given the large observed drop in albedo. Instead, the mass loss upon humidification suggests that at least 15% of the volume of AS seeds had been replaced under dry conditions by glyoxal reaction products that could break down into gas phase species once water was added. Simultaneous increases in gas-phase PTR-MS signals for $m/z$ 47 (formic acid) and 61 (acetic acid) indicate that these acids were two of the gas-phase products generated by humidification. Formic acid is a known byproduct of imidazole production by aqueous-phase glyoxal + ammonia reactions (De Haan et al., 2009a;Yu et al., 2011).

Dried seed aerosol particles atomized from AS-glycine mixtures were also exposed to 0.25 ppm glyoxal under dry conditions in experiment 2 (Figure S1). The response of these internally mixed seeds to glyoxal exposure was comparable to that of pure AS seeds. No growth was observed by SMPS, and aerosol albedo at 450 nm was anticorrelated with PTR-MS glyoxal signals at $m/z$ 59, as before. The most significant difference between the experiments is that there was no 15% loss of aerosol mass observed by SMPS upon humidification of the chamber to 50% RH, even though acetic and formic acid were again released into the gas phase. This may be due to the lower volatility of deprotonated seed particle materials (glycine vs. ammonia). In addition, most glycine-derivatized imidazole products have permanent positive charges and are not in equilibrium with volatile neutral forms (De Haan et al., 2009a).

To better understand the reactive processes happening in the dry aerosol particles, further experiments were conducted in a 300 L Tedlar chamber probed by Q-AMS, SMPS, CAPS-ssa, and CRD/PAS. Figure S2 shows an AMS ion correlation plot comparing average signals before and after 40 ppb glyoxal was added over a period of 25 minutes to the dry chamber containing AS aerosol in experiment 3. Unsurprisingly, the slow addition of this smaller amount of glyoxal did not cause observable net particle growth or a decline in aerosol albedo at 450 nm. A marginal (0.9 μg/m$^3$, S/N = 1.6) increase in total organic aerosol was observed by Q-AMS during glyoxal addition, associated with significant increases (+30% or more relative to conserved aerosol species) in ion signals at $m/z$ 15 (CH$_3^+$ or NH$^+$ fragments), 23 (Na$^+$), 29 (CHO$^+$ fragment), 47 (formic acid+H$^+$ or a CH$_3$O$_2^+$ fragment), 69 (imidazole-H$^+$), 81 (pyrazine-H$^+$), 97 and 119 (imidazole carboxyaldehyde "IC" +H$^+$ and Na$^+$, respectively). Slight decreases in aerosol water signals were observed at $m/z$ 16. The detection of particle-phase imidazole, pyrazine, and IC suggests the presence of larger, light-absorbing molecules such as 2,2'-biimidazole and N-heterocycle derivatives that are typically associated with the products detected here (Kampf et al., 2012;Hawkins et al., 2018;Grace et al., 2019). However, detection of the larger product molecules directly from dry aerosol may require a soft, direct ionization technique such as extractive electrospray ionization (EESI-) MS.

Proposed chemical mechanisms for brown carbon production at AS particle surfaces are summarized in Schemes 1 and S1. Except for steps where new N-heterocyclic rings are formed, all processes are reversible (Kampf et al., 2012). Thus, a reduction of gas-phase glyoxal concentrations will shift reversible reactions away from BrC back towards simple N-heterocycle products, which do not absorb 450 nm light. Humidification to 50% RH accelerates this shift by removing more glyoxal from the gas phase, and perhaps also by hydrolysis of double bonds and dilution effects (Rincón et al., 2010;Phillips and Smith, 2014, 2015). Humidification also triggers the observed evaporation of formate as formic acid, and perhaps the evaporation of other small N-containing products.

The anticorrelation of albedo with glyoxal concentrations in experiments 1-4 is summarized in Figure 2. Although AS-glycine-glyoxal bulk aqueous mixtures have been shown to brown more than mixtures without glycine (Trainic et al., 2012;Powelson

et al., 2014), here we see that dry AS and AS/glycine aerosol particles brown similarly for a given concentration of gas-phase glyoxal. This may indicate that glycine is not at the aerosol surface, or that glycine surfaces, when present, are less able to retain adsorbed water in the dry chamber. We therefore fit the combined dataset from all 4 experiments. Albedo shows a clear downward curvature at high glyoxal concentrations, such that the relationship is best fit by a $2^{nd}$ order polynomial. This suggests that the formation of the compounds absorbing at 450 nm is proportional to $[glyoxal]^2$. While glyoxal + ammonium reactions are $1^{st}$ order in glyoxal in dilute solution when $[glyoxal] \times [NH_4^+] < 1.2$ M, they switch to $2^{nd}$ order at higher concentrations (Noziere et al., 2009), which are likely in these dry experiments.

Reversible surface browning of AS aerosol under dry conditions was also recently observed during exposures to methylglyoxal gas (De Haan et al., 2017). The albedo values observed before and after two methylglyoxal additions are shown for comparison in Figure 2. Although the data shows a slight negative offset due to particle size effects, judging by the slope it is clear that methylglyoxal's effect on the albedo of dry AS aerosol is significantly less than glyoxal. This is the opposite of the trend in brown carbon production in bulk aqueous solutions at pH 5, where methylglyoxal is much more effective in generating light-absorbing products (Powelson et al., 2014), perhaps due to the fact that its ketone functional group is far less likely to be inactivated by hydration than the aldehyde groups on both molecules. However, in these dry aerosol experiments where water is scarce, glyoxal's greater attraction to water (seen in its much higher Henry's law coefficient) (Betterton and Hoffmann, 1988;Ip et al., 2009;Kampf et al., 2013) may allow it to interact with small amounts of adsorbed water at the AS aerosol surface far more effectively than methylglyoxal.

**3.2 Dry MeAS or sodium sulfate aerosol (experiments 5 - 6)**

Gas phase glyoxal was added to a few other types of seed particles in the small chamber. In experiments on MeAS seeds (expt. 5, Figure 3 top panel), the slow addition of 140 ppb of glyoxal caused a matching drop of -0.15 in aerosol albedo measured by CAPS-ssa at 450 nm and an increase in aerosol absorbance to 28 Mm$^{-1}$ measured by PAS at 405 nm. The albedo decline at 450 nm (Figure 2, green circles) is 4× greater than observed on AS seeds at similar glyoxal concentrations. This result is consistent with earlier aqueous-phase studies showing greater browning in glyoxal – methylamine mixtures than in glyoxal – AS mixtures at the same pH (Powelson et al., 2014).

Albedo at 405 nm was calculated from PAS and CRD signals in expt. 5 (Figure S3), showing that albedo had dropped to 0.30 by 1:11 pm and remained at this level for 45 minutes. These albedo values indicate that maximum light absorbance at 405 nm was 4.7× greater than at 450 nm, and persisted for a longer period of time after glyoxal gas concentrations decline. At even longer wavelengths (530 nm), PAS aerosol absorbance reached only 0.9 Mm$^{-1}$, further indicating highly wavelength-dependent light absorption. The absorption spectra of atmospheric brown carbon are typically well-fit by exponential decay functions. Such featureless spectra can be characterized by an Ångstrom absorption coefficient $\alpha$, which is the slope of a log(absorbance)

vs log(wavelength) plot. Comparing the amount of light absorbance at 405, 450, and 530 nm at 1:11 pm in experiment 5 gives
$\alpha$ = -12.7 ±0.8 (Figure S5). A similar analysis of expt. 4 (Figures S6 and S7, dry AS + glyoxal), the only other experiment
with measurable absorbance at all 3 wavelengths, gives a comparable $\alpha$ = -16 ±4. Thus, it appears that brown carbon formed
by glyoxal under dry conditions on AS and MeAS aerosol absorb light with similar wavelength dependence.

In an experiment on dry sodium sulfate seeds at ~35% RH in the small Tedlar chamber (expt. 6, Figure 3) glyoxal
concentrations were increased from zero to ~2000 ppb over 30 min. During this time, albedo at 450 nm remained at 1 ±0.005,
and no aerosol absorbance was measured by PAS at either 530 or 405 nm. The lack of browning observed even at such high
glyoxal concentrations confirms that ammonium or methylammonium ions (or ammonia or methylamine) are necessary
reaction partners with glyoxal in the browning process observed in experiments 1-5. It also confirms that our CAPS and PAS
measurements are not biased by absorbance due to gas-phase glyoxal.

### 3.3 Deliquesced AS aerosol (experiments 7-9)

Finally, three experiments exploring browning on wet rather than dry AS aerosol were conducted at RH ranging from 38 to
81%. The highest humidity experiment (expt. 7) is summarized in Figure S8. In experiments 7–9, albedo declines of 0.013
or less were observed following addition of 1.1, 1.2, and 0.12 ppm of glyoxal gas, respectively. If graphed in Figure 2, the
resulting slopes for experiments 7-8 would be more than 1000x flatter than the methylglyoxal data shown for comparison.
While some of the glyoxal gas added may have been quickly lost to the walls of the humid chambers as an equilibrium is
established (Kroll et al., 2005), especially in experiments 7 and 9, it is clear that wet AS aerosol particles brown much less
than dry AS, AS/glycine, or MeAS aerosol upon exposure to glyoxal.

Enhanced AS aerosol browning under dry conditions is surprising, given that glyoxal Maillard chemistry is normally
considered an aqueous-phase process. One clue to the nature of the dry browning process is seen in the slight depletion of
water signals observed in all dry experiments probed by Q-AMS (#3, 4, 5, Figures S2 and S9) after browning caused by glyoxal
exposure. (Most water is removed from aerosol particles in the AMS inlet.) The extra water depletion associated with glyoxal
exposure of dry aerosol, which was not observed in deliquesced aerosol experiments probed by Q-AMS (#7-8), suggests that
even under dry conditions, glyoxal is able to access and deplete trace amounts of aerosol-phase surface water. Any adsorbed
water would be saturated with ammonium (or methylammonium) sulfate, and the presence of dissolved AS is known to greatly
increase glyoxal uptake via a "salting in" effect (Kampf et al., 2013;Waxman et al., 2015), while methylglyoxal solubility is
reduced by salting out (Waxman et al., 2015). Thus, both glyoxal and AS are expected to be concentrated in any surface-
adsorbed water present. In a previous study, similar reasoning was used to explain glyoxal uptake on solid seed particles at
RH levels as low as 10% (Corrigan et al., 2008). Furthermore, the scarcity of water will favor dehydration of products, helping
to form light-absorbing conjugated double bonds.

## 4 Discussion

Since methylglyoxal is generally less abundant in the atmosphere than glyoxal (Igawa et al., 1989;Munger et al., 1995;Matsumoto et al., 2005), and since the browning of dry AS by methylglyoxal is much less than that of glyoxal likely due to salting effects (Kampf et al., 2013;Waxman et al., 2015), we will focus on the effects of instantaneous browning of atmospheric aerosol particles due to interaction with glyoxal. We will assume that all tropospheric sulfate particles contain ammonium, and, as an upper limit, that solid-phase tropospheric sulfate particles would brown as much as the pure, fully dry AS particles used in this study regardless of the presence of additional aerosol species. The first assumption is generally reasonable (Jimenez et al., 2009), since acidic sulfate aerosol takes up ammonia in the atmosphere, while the second assumption will clearly result in the estimation of an upper limit, since the presence of other materials at aerosol particle surfaces has been shown to limit the extent of the interactions between glyoxal and ammonium ions (Drozd and McNeill, 2014), and few locations in the troposphere are as dry as in this study. Tropospheric aerosol particles are typically semi-solid or solid phase, except low over the Amazon and Arctic (Shiraiwa et al., 2017).

Using the function *albedo = -0.97[GX]² - 0.16[GX] + 1.00* from Figure 2, a global 24-h average glyoxal concentration of ~70 ppt (Fu et al., 2008;Zhou and Mopper, 1990;Munger et al., 1995;Spaulding et al., 2003;Matsunaga et al., 2004;Müller et al., 2005;Ieda et al., 2006) would lower particle albedo at 450 nm ($\Delta Albedo(450)$) by only $1.1 \times 10^{-5}$. Using our measured Ångstrom absorption coefficient $\alpha$ = -16 for glyoxal + AS brown carbon formed under dry conditions, we estimated albedo depression at other wavelengths ($\Delta Albedo(\lambda)$) between 280 and 4000 nm using Eq. (1):

$$\frac{\log\left(\dfrac{\Delta Albedo(\lambda)}{\Delta Albedo(450)}\right)}{\log\left(\dfrac{\lambda}{450}\right)} = \alpha$$

(1)

These albedo decreases were multiplied by the solar spectrum (ASTM G173-03) times the wavelength-dependent scattering function of AS aerosol (Nemesure et al., 1995) at each wavelength (Figure S10), and then integrated across the spectrum. 97% of the solar energy absorbed by this brown carbon source is predicted to be in the UV range, with the absorbed energy peaking near 330 nm. The total fraction of energy absorbed by glyoxal + AS brown carbon (absorption × solar spectrum × scattering function) is calculated to be $1.9 \times 10^{-4}$, relative to the total energy scattered by AS aerosol (solar spectrum × scattering function). We then multiply this energy fraction times the magnitude of global direct radiative forcing due to sulfate scattering, estimated by the IPCC to be -0.4 ±0.2 W/m$^2$ (Ramaswamy et al., 2018), to quantify a global radiative forcing of $+7.6 \times 10^{-5}$ W/m$^2$ by dry browning of ammonium sulfate aerosol. This climate forcing is negligible compared to the global net aerosol direct effect (-0.45 ±0.5 W/m$^2$) or absorption by black carbon ($+0.4$ $^{+0.4}_{-0.35}$ W/m$^2$) (Ramaswamy et al., 2018), and is less than 1% of estimates of radiative forcing by secondary brown carbon (+0.015 to +0.081 W/m$^2$) (Mukai and Ambe, 1986;Hecobian et al.,

2010;Shamjad et al., 2015;Tuccella et al., 2020). While dry browning of ammonium sulfate aerosol in the presence of ambient glyoxal thus does not appear to be globally significant in terms of radiative forcing, it may be regionally significant in polluted areas where glyoxal concentrations can greatly exceed 70 ppt (Volkamer et al., 2005a), where larger loadings of AS aerosol are present, or where aerosol browning by glyoxal occurs in the upper troposphere (Zhang et al., 2017).

## 5 Supporting Information

Supporting Information is available: proposed reaction scheme including humidification, data summaries of experiments 2, 4, 7, 8, and 9, Ångstrom coefficient plots for experiments 4 and 8, Q-AMS plots summarizing the effects of glyoxal addition in experiments 3 and 8, and estimated spectrum of absorbance of sulfate-scattered solar radiation due to glyoxal uptake.

## 6 Acknowledgments

This work was funded by NSF grant AGS-1523178 and AGS-1826593. L. N. Hawkins was funded by the Barbara Stokes Dewey Foundation and Research Corporation (CCSA 22473). The authors thank Mila Ródenas García (CEAM) for access to the *Main Polwin* MATLAB program and for glyoxal FTIR reference spectra. CNRS-INSU is gratefully acknowledged for supporting CESAM as an open facility through the National Instrument label. The CESAM chamber has received funding from the European Union's Horizon 2020 research and innovation program through the EUROCHAMP-2020 Infrastructure Activity under grant agreement No 730997.

## Data availability

Data is available from the authors upon request.

## Author Contributions

David De Haan guided the project and wrote the manuscript. Lelia Hawkins and Jean-François Doussin guided large chamber experiments. Margaret Tolbert guided small chamber experiments. Kevin Jansen conducted small chamber experiments. Hannah Welsh, Raunak Pednekar, Alexia de Loera, Natalie Jimenez, Mathieu Cazaunau, and Edouard Pangui conducted large chamber experiments. Aline Gratien and Antonin Bergé quantified glyoxal by FTIR in the large chamber. Paola Formenti provided assistance in interpreting optical measurements.

## Competing Interests

The authors declare no competing interests.

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

**Table 1: Summary of Glyoxal Gas Addition Experiments:**

| Expt # | [GX][a] (ppm) | aerosol type | seed aerosol conc. ($\mu g/m^3$) | seed density ($g/cm^3$) | RH at glyoxal addition (%) | mass increase ($\mu g/m^3$) | albedo change, 450 nm |
|---|---|---|---|---|---|---|---|
| 1a | 0.05 | AS | 145 | 1.77 | < 5 | < 0.3 | -0.034[b] |
| 1b | 0.50 | AS | 145 | 1.77 | < 5 | -3 (decr) | -0.233[b] |
| 2 | 0.25 | AS/Gly | 100 | 1.30 | < 5 | < 1 | -0.094[b] |
| 3[c] | 0.04[d] | AS | 90 | 1.77 | < 5 | 0.9[e,f] | -0.001 |
| 4[c] | 0.30[d] | AS | 40[f] | 1.77 | < 5 | < 1 | -0.069 |
| 5[c] | 0.14[d] | MeAS | 80[f] | 1.44[g] | < 5 | < 1 | -0.15 |
| 6[c] | 2.0[d] | $Na_2SO_4$ | 70[f] | 1.46[h] | 35 | < 1 | 0 |
| 7[c] | 1.1[d] | wet AS | 180[f] | 1.24[i] | 81 | < 1 | -0.012 |
| 8[c] | ~1.2[d] | wet AS | 170 | 1.43[i] | 38 | < 1 | -0.010 |
| 9 | 0.12[j] | wet AS | 90 | 1.26[i] | 77 | < 1.6 | -0.013[b] |

Notes: GX = glyoxal; AS = ammonium sulfate; Gly = glycine; MeAS = methylammonium sulfate. **a**: Tabulated GX concentrations are peak values measured by PTR-MS, ±20% rel. uncertainty unless otherwise stated. **b**: occurring within 5 min. of GX pulse addition. **c**: Experiment performed in 300 L Tedlar bag. **d**: GX added gradually rather than in pulse; max. concentration estimated from PAS measurements at chamber inlet. **e**: Organic aerosol growth. **f**: measured by Q-AMS. **g**: from (Qiu and Zhang, 2012). **h**: from (Merck, 1983), for the decahydrate. **i**: from AIM model IV (Clegg and Wexler, 2011) **j**: Estimated from addition bulb pressure and comparison of PTR-MS signals of *m/z* 72 imine product.

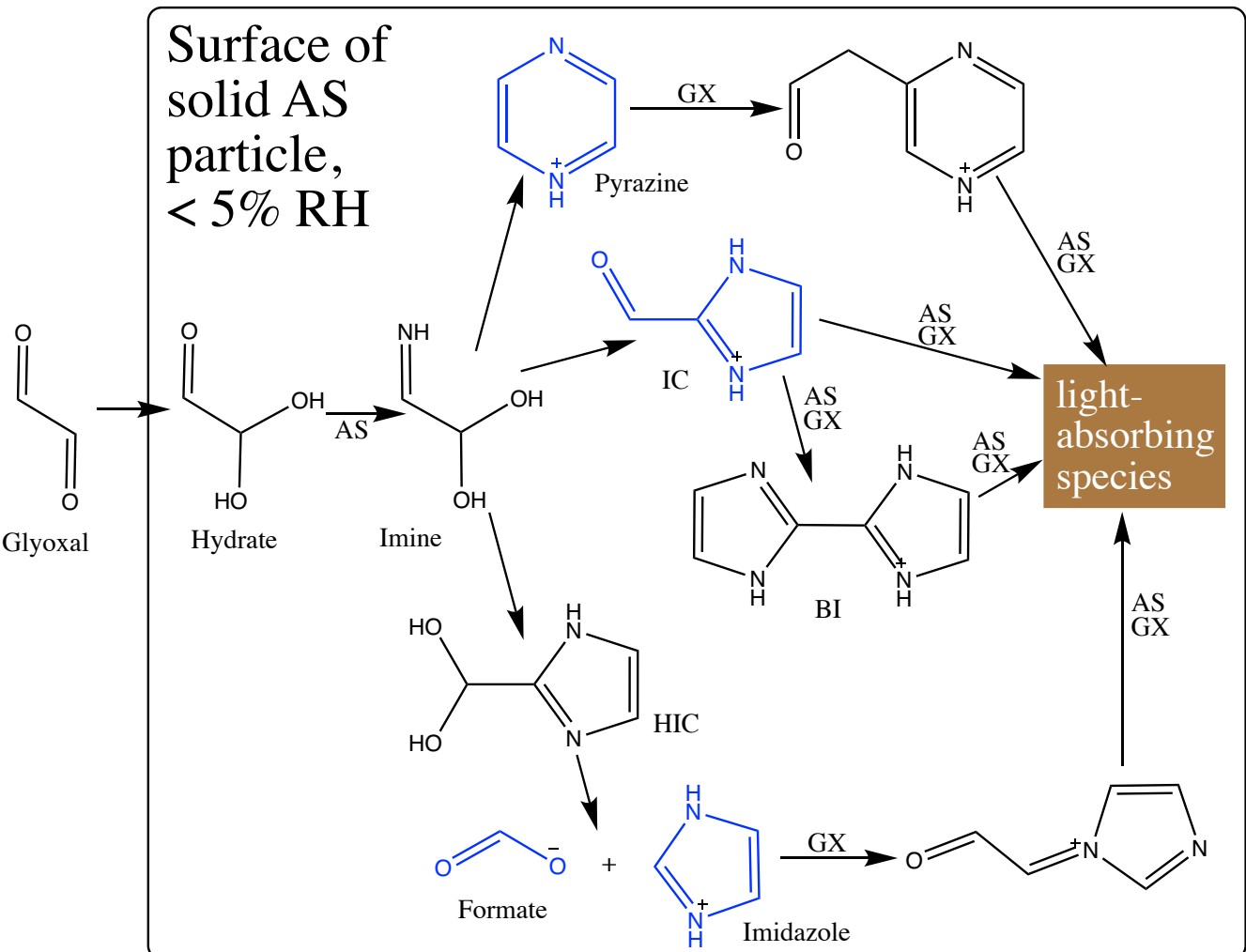


**Scheme 1:** Proposed brown carbon formation pathways of glyoxal reacting at solid AS aerosol particle surfaces. Products detected in this study are shown in blue. IC = 1H-imidazole-2-carboxaldehyde. BI = 2,2'-biimidazole (Kampf et al., 2012). AS = ammonium sulfate. GX = glyoxal. We assume, following (Kampf et al., 2012), that all reactions are reversible except for formation of N-heterocycle rings. See Scheme S1 for corresponding diagram of pathways under conditions of

humidification.

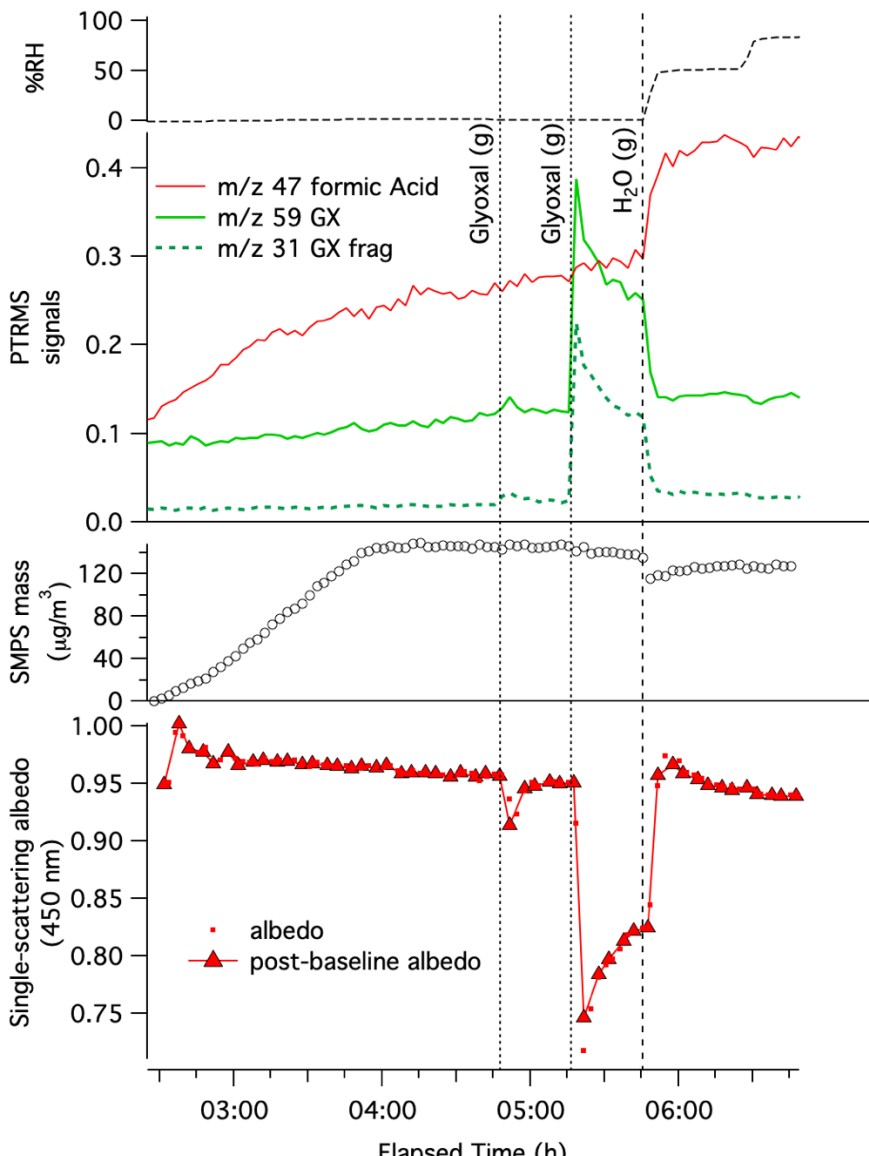

**Figure 1**: Pulse glyoxal addition experiment 1 on dry AS aerosol in CESAM chamber. Top: chamber RH. Middle panels: Dilution- and water-corrected PTR-MS traces for gas-phase glyoxal ($m/z$ = 59, green line), a glyoxal fragment ($m/z$ 31, dark green dotted line), formic acid ($m/z$ 47, red line); SMPS particulate mass corrected for wall losses and dilution (assuming aerosol density = 1.77 g/cm$^3$, open black circles), with increasing mass for first 90 minutes indicating AS aerosol addition to chamber. Bottom: single-scattering albedo (red dots), and albedo values calculated from data immediately following instrument baseline on gas-phase contents of chamber (red triangles), measured by CAPS-ssa at 450 nm. Sequential gas "(g)" additions of 0.05 and 0.50 ppm glyoxal (vertical dotted line) and water vapor addition (dashed lines) are labeled. Elapsed time is measured from start of N$_2$ addition to the evacuated chamber.

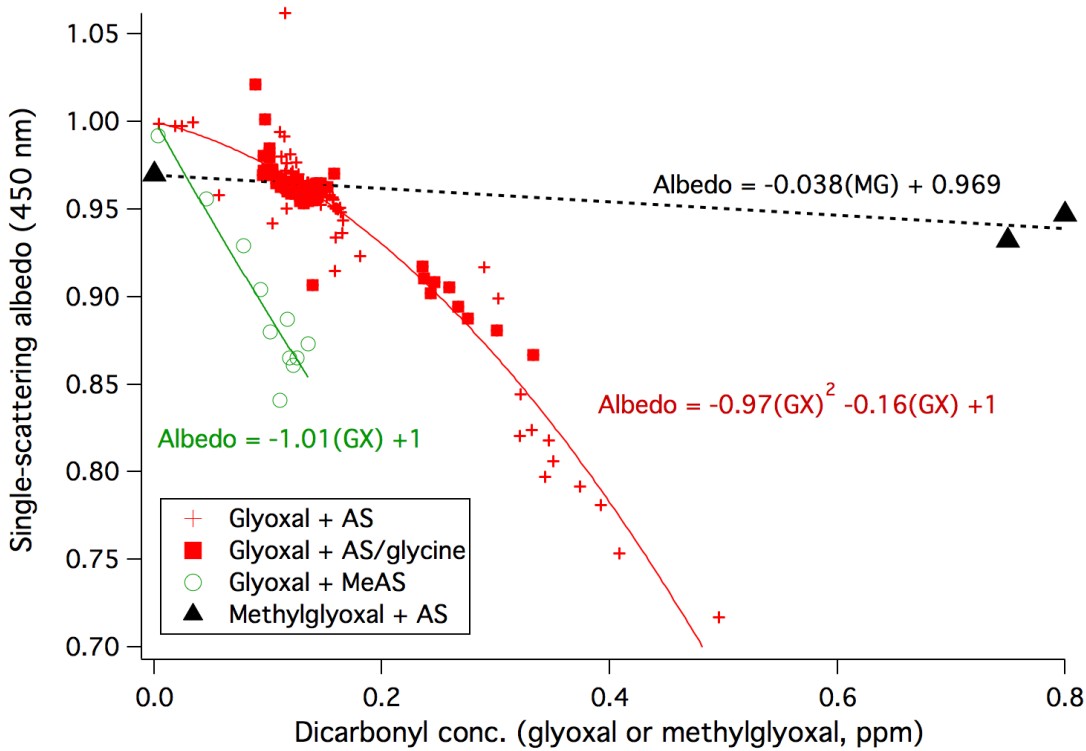

**Figure 2**: Top: Anticorrelation of particle single-scattering albedo at 450 nm (with a 3 – 7 min delay) with gas-phase concentrations of glyoxal (Expts. 1, 3, and 4: dry AS, red +; Expt. 2: dry AS/glycine, filled red squares; Expt. 5: dry MeAS, green circles) and methylglyoxal (black triangles, from (De Haan et al., 2017)) as measured by PTR-MS (expts. 1-2 and methylglyoxal data) or photoacoustic spectroscopy (expts. 3-5).

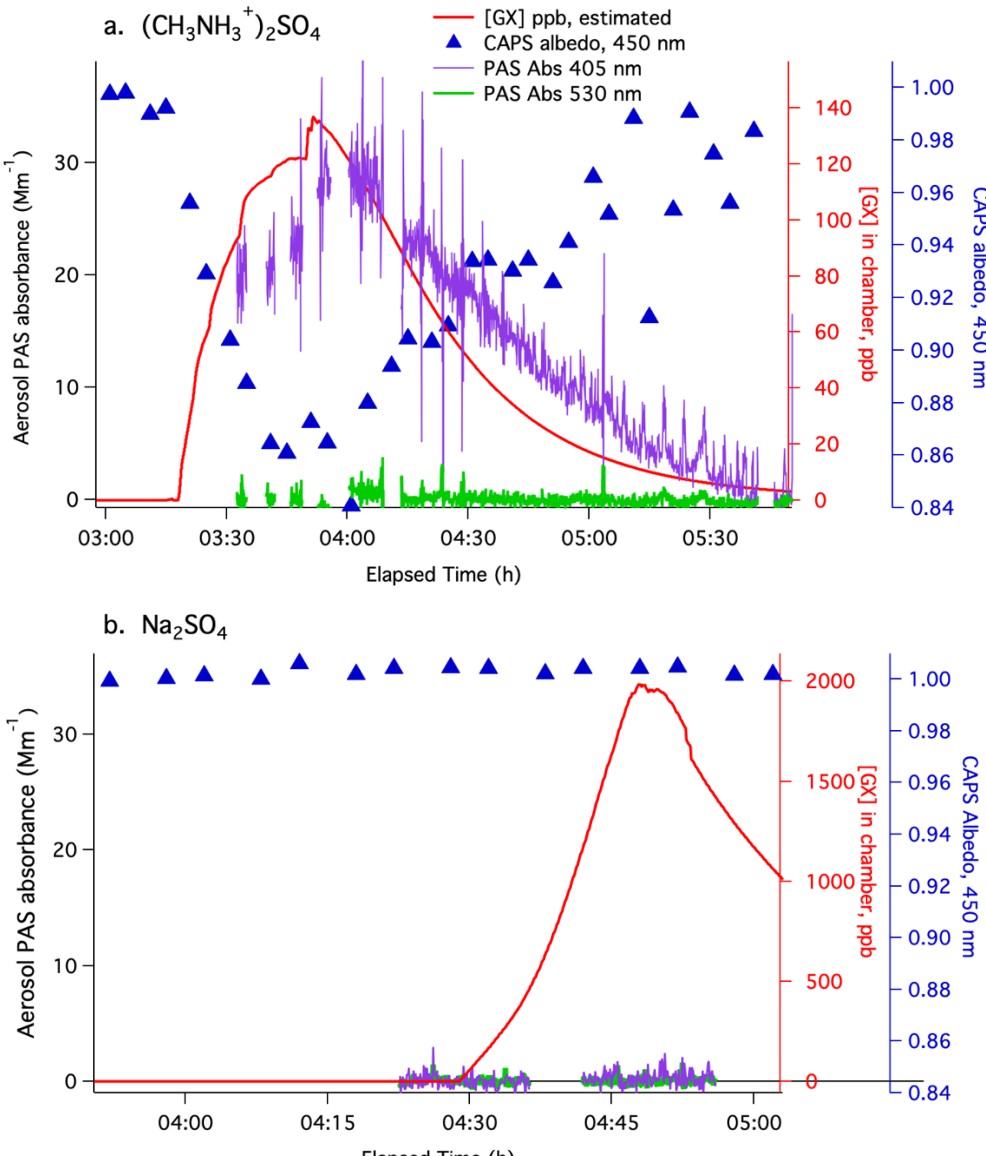

**Figure 3**: Gradual glyoxal addition experiments in small Tedlar chamber on **a.)** dry methylammonium sulfate aerosol (experiment 5), and **b.)** dry sodium sulfate aerosol (experiment 6). Aerosol absorbance measured by PAS at 405 nm (purple lines) and 530 nm (green lines), aerosol albedo measured by CAPS at 450 nm (blue triangles, blue axis), and estimated glyoxal concentrations in the chamber (red line, red axis, calculated using GX measurements at inlet, flow mixing ratios, and GX wall loss rate = $6.7 \times 10^{-4}$ s$^{-1}$). CRD, SMPS, and 405 nm albedo data for these experiments is displayed in Figures S3 and S4, respectively.