# Peer review of "Glyoxal's impact on dry ammonium salts: fast and reversible surface aerosol browning"

_Atmospheric Chemistry and Physics, 2020_

## Referee Comment (RC1) · Anonymous Referee #1 · 31 Mar 2020

In this very well written manuscript the authors discuss results of thoroughly planned and described chamber experiments on the browning of aerosol particles upon exposure to gaseous glyoxal. Seed aerosols consisted of ammonium sulfate, methylammonium sulfate, mixed ammonium/glycine sulfate, and sodium sulfate, and were exposed to glyoxal under different relative humidity conditions. The authors observe a 'reversible' browning of the aerosol particles, when two conditions are met: 1) dry chamber conditions (<5% RH), and 2) amine functionalities are present in seed particles. Furthermore, the authors try to quantify the contribution of the observed browning to global radiative forcing induced by secondary brown carbon and conclude that its contribution is negligible with <1%. The findings are discussed very well and contribute significantly to the field. I have a few comments beside technical corrections that need

to be addressed in a minor revision prior publication:

General comments:

1) The observation that the browning process appears to be reversible is discussed at several points throughout the manuscript. Indeed, the single-scattering albedo in Figure 1 starts to recover with declining glyoxal gas-phase concentration. However, the addition of water vapor results in a sudden recovery of the albedo back to the baseline. This effect could also be due to a dilution effect, as the absorption properties of glyoxal derived brown carbon is known to be highly concentration dependent. Could the authors comment on a potential dilution effect, especially of surface active brown carbon constituents?

2) Non-reversibly formed brown carbon from glyoxal might anyways not be present in appreciable amounts, given the timescale available for their formation. Maybe it would benefit the paper to expand a corresponding discussion a little bit, e.g., in line 201, where the formation of light-absorbing double bonds is mentioned.

3) The control experiment utilizing sodium sulfate seed aerosol is very helpful. It not only confirms the involvement of amine species in the observed fast and reversible browning effect, but also that the CAPS and PAS measurements were not biased by gas-phase glyoxal. Which other measures have been taken to ensure artifact-free analysis?

Technical comments:

4) Figure 3: Could the x-axis be labeled with "Elapsed Time (h)" as in Figure 1?

5) Line 91: Brackets could be removed.

Comments on the Supporting Information:

6) Figure S1: Start of chamber illumination (red line) – Did I miss it or is not there?

7) Figure S2: Please include axis labels; the molecular masses for m/z 119 in brackets should be 96+23; Is it common for AMS analysis that formic acid appears as a protonated species at m/z 47?

8) Figure S3: Exponents of the wall loss rate should be superscript

9) Figures S3, S4, S6: Could the x-axis label be changed to "Elapsed Time (h)"?

---

## Referee Comment (RC2) · Anonymous Referee #2 · 19 Apr 2020

Summary and recommendation:

In this study, De Haan et al. report on fast browning of ammonium sulfate (AS), AS/glycine and methylammonium sulfate (MeAS) aerosol particles under dry conditions when exposed to gas phase glyoxal. It is shown that this browning process is not accompanied by noticeable particle growth and that this is reversed when water vapor is added. Remarkably, dry methylammonium sulfate aerosol was found to brown 4 times more than dry AS aerosol, and deliquesced AS aerosol browns much less than dry AS aerosol. Lastly, the authors estimate the impact of these browning processes on global radiative forcing, concluding that its contribution might only be important on a regional scale under polluted conditions.

The authors acquired a nice dataset with state-of-the-art instruments during their

chamber experiments, which will contribute to our understanding of brown carbon formation in aerosol particles. However, I see several points in the manuscript, which need to be addressed before I can recommend its publication in ACP (as detailed below).

Major comments:

1) Figure 1 / L111f: I cannot follow the authors' explanation of the SMPS mass loss upon chamber humidification. There is no mass increase upon GX addition under low RH levels, indicating that GX uptake to particles should be rather low. This is also consistent with the authors' assumption that most GX goes to the walls during this time (L102). So, how can a mass loss of ∼15% upon humidification then be explained by destruction of light-absorbing products formed through GX uptake? If there was only minimal uptake of GX, the particles should still consist mainly of AS. One factor that might be important here is the wall loss correction, as particle wall losses can significantly change upon RH increase. Did the authors consider this in their analysis?

2) I think the manuscript would largely benefit from some reaction mechanisms. This would help non-expert readers a lot in grasping the chemistry behind the browning reactions / imidazole formation. In the current version, all details on known and hypothesized chemical pathways are rather "hidden" in the text.

3) The discussion section is extremely speculative. Especially, since RH conditions of <5% are commonly far from tropospheric conditions. The mere statement that "tropospheric aerosol is nevertheless typically semi-solid or solid" (Line 212) does not eliminate this constraint. Therefore, I find it difficult to agree on the authors conclusion that aerosol browning by AS + GX under dry conditions may be regionally important.

Specific comments:

1) L67 and L93f: How was the wall loss correction performed? It would be helpful to have some more details here.

2a) Figure 1 / L90: I would suggest to set t = 0 h for the GX addition to improve

readability.

2b) Figure 1: unit of aerosol density is missing in the caption

3) L124 / Figure S1: The strong difference between the experiments with AS and AS/glycine seeds needs a more detailed discussion. Only referring to differences in volatility without further discussion seems too vague.

4) Table 1 and corresponding figures: It is not reasonable to assume a density of 1 g/cm3 for the seed particles. Please use some more realistic numbers.

5) Figure S2: Axis labels are missing.

6) Line 129f: Are these increases in AMS ion signals significant? Would it be possible to give some more quantitative information here? Otherwise, it is not possible to judge whether this is an important contribution to the composition of the particles. Furthermore, is the resolution of a Q-AMS sufficient to assign all these signals unambiguously to certain compounds and corresponding fragments?

7) Why are experiments 8 and 9 discussed before experiment 5–7? I would suggest to rename the experiments or to reorganize the manuscript.

8) Figures 3, S3, S4, and S6: I would suggest using a relative time on the x-axis instead of the time of the day. This would improve readability. Moreover, it would be consistent with Figure 1.

9) It would be nice to see at least for one of the experiments on wet AS an experimental overview figure (e.g., in the supplement) in the style of Fig. 1. Currently, the reader has to imagine how the experimental procedure and corresponding data looked like.

10) With a logarithmic y axis it should be possible to show both curves in one figure.

11) Table 1, experiment 1b: How did the authors infer a decrease of 3 $\mu$g/m3? From Fig. 1 the decrease seems to be in the range of 8–10 $\mu$g/m3. Moreover, this is inconsistent with the authors' statement (L111), that a mass decrease of $\sim$15% was observed.

Technical comments:

1) L95: I guess this should read "lack of uptake".

2) Caption of Table 1: The $\pm$ sign should be formatted in black.

---

## Referee Comment (RC3) · Anonymous Referee #3 · 19 Apr 2020

Summary and recommendation:

This paper details experiments the production of brown-carbon formation and light-scattering ability as a function of relative humidity and minor chemical differences in aerosol type. The authors find that dry particles produce larger albedo changes upon introduction of glyoxal into their chamber than for wet particles. They also connect relative humidity changes to albedo and particle mass measurements and explain these results in the context of global radiative forcing impact, which may hold local significance even if not globally significant. In general, the results are clearly presented, the conclusions well thought out, and offer meaningful contributions to the field at large. I believe there are, however, several points that the authors need to clarify before acceptance for publication to better articulate the impact of the work.

[Figure]

Major comments:

1. In Table 1 is appears that reactions 1a and 3 are nearly identical with the exception of chamber. The results, as given by aerosol concentration, mass increase, and albedo change, however, and markedly different. Is the only difference here the chamber used? It is not clear from the text how the reader should understand these differing results. The authors would do to clarify how the two chambers affect their results and offer some guidance on how their results should be read in light of those differences.

2. In lines 95-99 the authors make the case that the lack of particle growth is consistent with a lack of uptake of glyoxal at low RH, but in lines 195-196, the possible (though small) uptake of glyoxal is highlighted as a reason for water depletion. If the results suggest that glyoxal is able to access surface water (and thus uptake to the particle even under dry conditions), shouldn't this be taken into account in the particle growth discussion?

3. Following on to point #2, in lines 102-104 the discussion is hard to follow. The loss of glyoxal is largely to the steel walls, and yet the large albedo decline is also due to glyoxal-driven surface reactions? Does this suggest that even minor amounts of reactivity lead to very high albedo changes? I think the confusion readers will have with this section that the authors need to clear up is related to the sizing language. How should we read "largely", "large", and "at least some" in order to understand the weight of the argument the authors are making here?

4. In regards to Figure 1, how much of the recovery of albedo upon introduction of water is due to exclusively to the introduction of the water and how much is due to there still being glyoxal in the chamber as evidenced by the PTR-MS signals at 59 and 31? Unlike the earlier addition of a small glyoxal concentration, this larger addition was not allowed to return to baseline, and while this may not matter in the resultant albedo recovery upon water addition discussion, this overlap isn't addressed adequately in the text. Essentially, is the albedo increase due to the addition of the water, or the resultant

dilution/loss of glyoxal signal?

5. In lines 143-144, and Figure 2 the case is made for albedo change as a function of glyoxal concentration being a second order polynomial with respect to glyoxal concentration. Is there a physical explanation that would help defend this choice, or is it simply what fit the data? Or, is there a reason that using a two-slope approach (say a linear fit to the data at [glyoxal] < 0.35 ppm and another linear fit to the data at [glyoxal] > 0.35 ppm) wouldn't also successfully capture the data? The authors should offer with what significance the readers should approach this polynomial fit and [gly]^2 dependence.

Minor and technical comments:

1. At least when I downloaded a copy of the paper, Table 1 had some jumbled values in the last column (it would appear that those values are line numbers). The authors should check this Table to ensure that everything is in the place they expect it to be. This could very well be an artifact of the download and not the paper, and so this might disappear in final publication.

2. In regards to the discussion on lines 195-196 again, a more curiosity-driven question the authors may consider, if they wish, commenting on is: can this ability of glyoxal to access surface water lead to a localized area of highly concentrated glyoxal (and that would thus accelerate chemical reactivity)?

---

## Author Comment (AC1) · 10 Jun 2020

Reviewer 1
In this very well written manuscript the authors discuss results of thoroughly planned and described chamber experiments on the browning of aerosol particles upon exposure to gaseous glyoxal. Seed aerosols consisted of ammonium sulfate, methylammonium sulfate, mixed ammonium/glycine sulfate, and sodium sulfate, and were exposed to glyoxal under different relative humidity conditions. The authors observe a 'reversible' browning of the aerosol particles, when two conditions are met: 1) dry chamber conditions (<5% RH), and 2) amine functionalities are present in seed particles. Furthermore, the authors try to quantify the contribution of the observed browning to global radiative forcing induced by secondary brown carbon and conclude that its contribution is negligible with < 1%. The findings are discussed very well and contribute significantly to the field. I have a few comments beside technical corrections that need to be addressed in a minor revision prior publication:

General comments:
1) The observation that the browning process appears to be reversible is discussed at several points throughout the manuscript. Indeed, the single-scattering albedo in Figure 1 starts to recover with declining glyoxal gas-phase concentration. However, the addition of water vapor results in a sudden recovery of the albedo back to the baseline. This effect could also be due to a dilution effect, as the absorption properties of glyoxal derived brown carbon is known to be highly concentration dependent. Could the authors comment on a potential dilution effect, especially of surface active brown carbon constituents?

This comment is very thought-provoking. As noted by the reviewer, albedo declines when glyoxal gas concentrations go down, even before the humidity is increased. This dependence of particle absorptivity on glyoxal gas concentrations suggests full reversibility of brown carbon formation under dry conditions, and there is no reason to assume that this relationship would change after humidification. We cannot rule out the contribution of a dilution effect, however, and have re-written this section of the manuscript to include this possibility. In addition, we have included some information from the literature on the likely amount of water present on these effloresced aerosol particles at 50% RH – no more than 2 monolayers of adsorbed water are expected to be present.

"At $t$ = 5:46 h (Figure 1), the chamber was humidified to 50% RH, a level which would not deliquesce the AS seeds (Biskos et al., 2006) but which may produce as many as 1-2 monolayers of adsorbed water at aerosol surfaces consisting of solid AS (Denjean et al., 2014;Romakkaniemi et al., 2001) or glyoxal reaction products. (Hawkins et al., 2014). Humidification to 50% RH caused significant changes to both the gas and aerosol phases. Glyoxal PTR-MS signals and aerosol albedo returned within a few minutes back to near-baseline levels, while the dried aerosol mass measured by SMPS jumped downward by 15%. The loss of gas-phase glyoxal, again without aerosol growth, suggests that water greatly accelerated glyoxal loss rates to the steel chamber walls. The simultaneous albedo recovery and SMPS mass loss indicate that humidification destroyed all brown carbon products that absorb 450 nm light, converting some fraction of them to gas-phase products. A proposed mechanism for this process is discussed below. It is significant that no browning was observed in PILS-sampled aerosol at any

point during experiment 1, presumably due to the same mechanisms occurring during wet sampling."

A later paragraph describes the proposed mechanism:

"Proposed chemical mechanisms for brown carbon production at AS particle surfaces are summarized in Schemes 1 and S1.  Except for steps where new N-heterocyclic rings are formed, all processes are reversible. (Kampf et al., 2012)  Thus, a reduction of gas-phase glyoxal concentrations will shift reversible reactions away from BrC back towards simple N-heterocycle products, which do not absorb 450 nm light.  Humidification to 50% RH accelerates this shift by removing more glyoxal from the gas phase, and perhaps also by hydrolysis of double bonds and dilution effects (Rincón et al., 2010;Phillips and Smith, 2014, 2015)."

2) Non-reversibly formed brown carbon from glyoxal might anyways not be present in appreciable amounts, given the timescale available for their formation. Maybe it would benefit the paper to expand a corresponding discussion a little bit, e.g., in line 201, where the formation of light-absorbing double bonds is mentioned.

Imidazole ring formation in the glyoxal + AS system is thought to be irreversible (Kampf et al., 2012), so it is significant that we detect two imidazole ring products in the aerosol phase when glyoxal is added, and upon water addition we detect the release of formic acid to the gas phase. (Formic acid is irreversibly co-produced with imidazole as the C-C bond of glyoxal breaks.)  The formation of reversible and irreversible products, both detected and proposed, are now shown in a new Scheme 1.

3) The control experiment utilizing sodium sulfate seed aerosol is very helpful. It not only confirms the involvement of amine species in the observed fast and reversible browning effect, but also that the CAPS and PAS measurements were not biased by gas-phase glyoxal. Which other measures have been taken to ensure artifact-free analysis?

The CAPS instrument automatically samples through a filter periodically in order to remove signals from any gases present that absorb at 450 nm.  We increased the frequency of filter sampling corrections to every five minutes to minimize any potential effects from gas-phase absorbers.  This information is included in section 2.1.  PAS and CRD aerosol measurements were also periodically baselined through filters for similar reasons.  This information has been added to the manuscript, as shown below.  Periodic gaps in CRD and PAS data indicate when filter baselines were performed.  As noted by the reviewer, the control experiment demonstrates that these filter baselines successfully eliminated potential interference in aerosol optical signals even when high concentrations of glyoxal gas were present.

"Aerosol particles were sampled via diffusion driers by Q-AMS (Aerodyne), CAPS-ssa (Aerodyne, 450 nm), SMPS (TSI), CRD (530 nm) (Ugelow et al., 2017), and photoacoustic spectrometers (PAS, 405 and 530 nm) (Ugelow et al., 2017), all of which were periodically baselined through filters to eliminate interferences by gas-phase species. "

Technical comments:

4) Figure 3: Could the x-axis be labeled with "Elapsed Time (h)" as in Figure 1?

We have changed the x-axis of this figure to show elapsed time since start of experiment, consistent with Figure 1. We have changed the label as suggested.

5) Line 91: Brackets could be removed.

We have removed them.

Comments on the Supporting Information:

6) Figure S1: Start of chamber illumination (red line) – Did I miss it or is not there?

It's not there. We've removed reference to cloud events and chamber illumination from this figure, since neither are included.

7) Figure S2: Please include axis labels; the molecular masses for m/z 119 in brackets should be 96+23;

True. We have added the $H_+$ adduct mass to the caption, which makes the masses of both ions more self-explanatory, so we have removed the erroneous mass addition explanation.

Is it common for AMS analysis that formic acid appears as a protonated species at m/z 47?

The release of formic acid from aerosol upon humidification in our experiments was detected by PTR-MS, so we know that formic acid is one of the compounds being formed in the aerosol during dry glyoxal reactions. However, it has not been conclusively demonstrated that the Q-AMS $m/z$ 47 signal is in fact due to formic acid. This signal could also be due to a $CH_3O_{2+}$ fragment from hydrated imidazole carboxyaldehyde or a host of other known products. We have changed the text to read "formic acid or a $CH_3O_{2+}$ fragment".

8) Figure S3: Exponents of the wall loss rate should be superscript

We have fixed this formatting error.

9) Figures S3, S4, S6: Could the x-axis label be changed to "Elapsed Time (h)"?

We have made this change.

Reviewer 2

Summary and recommendation: In this study, De Haan et al. report on fast browning of ammonium sulfate (AS), AS/glycine and methylammonium sulfate (MeAS) aerosol particles under dry conditions when exposed to gas phase glyoxal. It is shown that this browning process is not accompanied by noticeable particle growth and that this is reversed when water vapor is added. Remarkably, dry methylammonium sulfate aerosol was found to brown 4 times more than dry AS aerosol, and deliquesced AS aerosol browns much less than dry AS aerosol. Lastly, the authors estimate the impact of these browning processes on global radiative forcing, concluding that its contribution might only be important on a regional scale under polluted conditions. The authors acquired a nice dataset with state-of-the-art instruments during their chamber experiments, which will contribute to our understanding of brown carbon formation in aerosol particles. However, I see several points in the manuscript, which need to be addressed before I can recommend its publication in ACP (as detailed below).

Major comments:

1) Figure 1 / L111f: I cannot follow the authors' explanation of the SMPS mass loss upon chamber humidification. There is no mass increase upon GX addition under low RH levels, indicating that GX uptake to particles should be rather low. This is also consistent with the authors' assumption that most GX goes to the walls during this time (L102). So, how can a mass loss of ~15% upon humidification then be explained by destruction of light-absorbing products formed through GX uptake? If there was only minimal uptake of GX, the particles should still consist mainly of AS. One factor that might be important here is the wall loss correction, as particle wall losses can significantly change upon RH increase. Did the authors consider this in their analysis?

A change in gas loss rates upon RH increase is likely, and this could cause loss of molecules from aerosol due to Henry's law equilibria. A change in physical particle wall loss rates with RH is unlikely, since all particle impacts with chamber walls are thought to be irreversible regardless of wetness. Even if both gas and particle loss rates were to increase with RH, however, this would cause a change in the slope of the SMPS mass data, not a downward jump as was observed. Instead, supported by 3 lines of evidence described below, we reason that 15% of the AS particle volume must have been converted by 30 min of reaction with glyoxal under dry conditions (with small amounts of adsorbed water) into reaction products of similar volume that could later break down into gas-phase species upon humidification. These evaporating gas-phase species include formic acid and acetic acid (whose release was detected by PTR-MS upon humidification in several large chamber experiments), and likely imidazole, ammonia, and a variety of imine species. Indeed, the aerosol-phase production of imidazole and other volatile C-N species was observed by Q-AMS during dry glyoxal exposure in several small chamber experiments, consistent with this explanation. In addition, the large changes in albedo upon glyoxal exposure also suggest that even though particle size was not changing, we should not equate this with a lack of glyoxal reactivity or uptake. The manuscript has been updated to make these arguments more clearly:

> "The 15% aerosol mass loss upon humidification to 50% RH is surprising, given that no corresponding mass gain was recorded during exposure to glyoxal under dry conditions. However, the lack of mass gain under dry conditions cannot be interpreted as a lack of glyoxal uptake or reactivity, given the large observed drop in albedo. Instead, the mass loss upon humidification suggests that at least 15% of the volume of AS seeds had been replaced under dry conditions by glyoxal reaction products that could break down into gas phase species once water was added. Simultaneous increases in gas-phase PTR-MS signals for $m/z$ 47 (formic acid) and 61 (acetic acid) indicate that these acids were two of the gas-phase products generated by humidification. Formic acid is a known byproduct of imidazole production by aqueous-phase glyoxal + ammonia reactions (De Haan et al., 2009;Yu et al., 2011)."

2) I think the manuscript would largely benefit from some reaction mechanisms. This would help non-expert readers a lot in grasping the chemistry behind the browning reactions / imidazole formation. In the current version, all details on known and hypothesized chemical pathways are rather "hidden" in the text.

We fully agree with this suggestion, and have added Schemes 1 and S1 to summarize known and proposed chemistry of this system.  We have placed it in the text at the point where we first discuss the AMS detection of imidazoles in the dry AS aerosol after glyoxal addition:

> "The detection of particle-phase imidazole, pyrazine, and IC suggests the presence of larger, light-absorbing molecules such as 2,2'-biimidazole and N-heterocycle derivatives that are typically associated with the products detected here. (Kampf et al., 2012;Hawkins et al., 2018;Grace et al., 2019)_However, detection of the larger product molecules directly from dry aerosol may require a soft, direct ionization technique such as extractive electrospray ionization (EESI-) MS.

> "Proposed chemical mechanisms for brown carbon production at AS particle surfaces are summarized in Schemes 1 and S1.  Except for steps where new N-heterocyclic rings are formed, all processes are reversible. (Kampf et al., 2012)  Thus, a reduction of gas-phase glyoxal concentrations will shift reversible reactions away from BrC back towards simple N-heterocycle products, which do not absorb 450 nm light.  Humidification to 50% RH accelerates this shift by removing more glyoxal from the gas phase, and perhaps also by hydrolysis of double bonds and dilution effects (Rincón et al., 2010;Phillips and Smith, 2014, 2015).  Humidification also triggers the observed evaporation of formate as formic acid, and perhaps the evaporation of other small N-containing products."

Here is Scheme 1:

[Figure]

**Scheme 1:** Proposed brown carbon formation pathways of glyoxal reacting at solid AS aerosol particle surfaces. Products detected in this study are shown in blue. IC = 1H-imidazole-2-carboxaldehyde. BI = 2,2'-biimidazole (Kampf et al., 2012). AS = ammonium sulfate. GX = glyoxal. We assume, following (Kampf et al., 2012), that all reactions are reversible except for formation of N-heterocycle rings. See Scheme S1 for corresponding diagram of pathways under conditions of humidification.

3) The discussion section is extremely speculative. Especially, since RH conditions of < 5% are commonly far from tropospheric conditions. The mere statement that "tropospheric aerosol is nevertheless typically semi-solid or solid" (Line 212) does not eliminate this constraint. Therefore, I find it difficult to agree on the authors conclusion that aerosol browning by AS + GX under dry conditions may be regionally important.

This difference between dry chamber conditions and the troposphere is one of two reasons our estimation of the atmospheric significance of browning of sulfate aerosol by glyoxal is an upper limit. We have now explicitly stated our assumption, made for the purposes of the estimation, that ammonium sulfate aerosol in our *dry* chamber react the same way as tropospheric sulfate aerosol, as long as they are *solid* phase. If we are wrong in this (now explicitly stated) assumption, our upper limit will be too high – still an upper limit, but less useful. Since our

estimated upper limit is already low enough that we can conclude that this browning process cannot be globally significant, the validity of this assumption does not impact the main conclusion of the paper. We have made the following revision to the text:

> "We will assume that all tropospheric sulfate particles contain ammonium, and, as an upper limit, that solid-phase tropospheric sulfate particles would brown as much as the pure, fully dry AS particles used in this study regardless of the presence of additional aerosol species. The first assumption is generally reasonable (Jimenez et al., 2009), since acidic sulfate aerosol takes up ammonia in the atmosphere, while the second assumption will clearly result in the estimation of an upper limit, since the presence of other materials at aerosol particle surfaces has been shown to limit the extent of the interactions between glyoxal and ammonium ions (Drozd and McNeill, 2014), and few locations in the troposphere are as dry as in this study. Tropospheric aerosol particles are typically semi-solid or solid phase, except low over the Amazon and Arctic (Shiraiwa et al., 2017)."

Specific comments:
1) L67 and L93f: How was the wall loss correction performed? It would be helpful to have some more details here.
Wall losses are calculated using size-dependent loss rates measured for AS test particles in the dry CESAM chamber. This information has been added to the manuscript in the Methods section 2.1:
> "SMPS size distributions were also corrected using size-dependent wall losses measured for AS particles in the chamber."

2) 2a) Figure 1 / L90: I would suggest to set t = 0 h for the GX addition to improve readability.
Regrettably, since the majority of experiments used gradual rather than pulsed glyoxal additions, and since experiment 1 used two glyoxal pulses, we would not be able to have a consistent time axis – as requested by the other two reviewers – if we used this suggestion.

2b) Figure 1: unit of aerosol density is missing in the caption
We have added the units, "g/cm$_3$", to the caption.

3) L124 / Figure S1: The strong difference between the experiments with AS and AS/glycine seeds needs a more detailed discussion. Only referring to differences in volatility without further discussion seems too vague.
We have added an additional line of reasoning to this section where we discuss the apparent lack of volatility of glyoxal + glycine reaction products upon water addition:
> "This may be due to the lower volatility of deprotonated seed particle materials (glycine vs. ammonia). In addition, most glycine-derivatized imidazole products have permanent positive charges and are not in equilibrium with volatile neutral forms (De Haan et al., 2009)."

4) Table 1 and corresponding figures: It is not reasonable to assume a density of 1 g/cm3 for the seed particles. Please use some more realistic numbers.

Once glyoxal reactions begin, particle densities are no longer certain. However, the densities of the aerosol seeds are known, and so we have added this information to Table 1. We have also corrected the seed aerosol concentrations listed in Table 1 that are based on SMPS measurements, which therefore depend on these densities. Since the seed aerosol concentrations are listed in Table 1 at the point of the start of glyoxal addition, the density cannot yet have changed, and so these numbers should be fully accurate. We have also corrected the SMPS masses in both figures that show such data (Figures 1 and S1) using seed particle densities, and note these densities in the captions of these figures.

5) Figure S2: Axis labels are missing.
We have added axis labels.

5) Line 129f: Are these increases in AMS ion signals significant? Would it be possible to give some more quantitative information here? Otherwise, it is not possible to judge whether this is an important contribution to the composition of the particles. Furthermore, is the resolution of a Q-AMS sufficient to assign all these signals unambiguously to certain compounds and corresponding fragments?

The increases described here are changes in averaged signals of at least +30% relative to conserved aerosol species such as sulfate ions, whose random variation when averaged signals are compared is always less than 10%. The data is shown visually in Figure S2. We have edited the manuscript to read:

> "A marginal (0.9 µg/m$^3$, S/N = 1.6) increase in total organic aerosol was observed by Q-AMS during glyoxal addition, associated with significant increases (+30% or more relative to conserved aerosol species) in ion signals at *m/z* 15 ($CH_3^+$ or $NH^+$ fragments), 23 ($Na^+$), 29 ($CHO^+$ fragment), 47 (formic acid+$H^+$ or a $CH_3O_2^+$ fragment), 69 (imidazole-$H^+$), 81 (pyrazine-$H^+$), 97 and 119 (imidazole carboxyaldehyde "IC" +$H^+$ and $Na^+$, respectively). Slight decreases in aerosol water signals were observed at *m/z* 16. "

Q-AMS does not permit unambiguous assignments. However, the glyoxal + AS system has been studied enough by high-resolution AMS that the identities of major peaks are known. We have changed the assignment of *m/z* 47 from "formic acid" to "formic acid+$H_+$ or a $CH_3O_{2+}$ fragment" throughout the manuscript and supplement, since this assignment is ambiguous.

7) Why are experiments 8 and 9 discussed before experiment 5–7? I would suggest to rename the experiments or to reorganize the manuscript.
We have renumbered the experiments so that they are now referred to in order.

8) Figures 3, S3, S4, and S6: I would suggest using a relative time on the x-axis instead of the time of the day. This would improve readability. Moreover, it would be consistent with Figure 1.
We have made this change to Figure 3. For the figures in the supplement, we have changed the time axis label to "Elapsed Time (h)"

9) It would be nice to see at least for one of the experiments on wet AS an experimental overview figure (e.g., in the supplement) in the style of Fig. 1. Currently, the reader has to imagine how the experimental procedure and corresponding data looked like.

We now show a new summary overview figure for experiment 7 (wet AS) in the SI section, Figure S8. This is the highest RH experiment, and shows that the albedo change of -0.012 upon glyoxal addition is easily measurable by CAPS at 450 nm.

10) With a logarithmic y axis it should be possible to show both curves in one figure.
I am not sure which curves and which figure this comment refers to.

11) Table 1, experiment 1b: How did the authors infer a decrease of 3 µg/m3? From Fig. 1 the decrease seems to be in the range of 8–10 µg/m3. Moreover, this is inconsistent with the authors' statement (L111), that a mass decrease of ~15% was observed.
The mass decrease of 3 µg/m3 listed in Table 1 for experiment 1b is the SMPS mass loss observed over 30 minutes after 0.50 ppm glyoxal was added. We have changed the Table 1 column label from "mass increase" to "mass increase upon glyoxal addition". The 8-10 µg/m3 mass loss in Figure 1 that the reviewer refers to is what was observed when the chamber was humidified to 50% RH. This value is not included in Table 1 because the chamber was rapidly humidified from <5 to 50% RH only in experiments 1b and 2.

Technical comments:
1)  L95: I guess this should read "lack of uptake".
        Corrected.
2) Caption of Table 1: The ± sign should be formatted in black.
        Corrected.

Reviewer 3

Summary and recommendation: This paper details experiments the production of brown-carbon formation and light scattering ability as a function of relative humidity and minor chemical differences in aerosol type. The authors find that dry particles produce larger albedo changes upon introduction of glyoxal into their chamber than for wet particles. They also connect relative humidity changes to albedo and particle mass measurements and explain these results in the context of global radiative forcing impact, which may hold local significance even if not globally significant. In general, the results are clearly presented, the conclusions well thought out, and offer meaningful contributions to the field at large. I believe there are, however, several points that the authors need to clarify before acceptance for publication to better articulate the impact of the work.

Major comments:
1. In Table 1 is appears that reactions 1a and 3 are nearly identical with the exception of chamber. The results, as given by aerosol concentration, mass increase, and albedo change, however, and markedly different. Is the only difference here the chamber used? It is not clear from the text how the reader should understand these differing results. The authors would do to clarify how the two chambers affect their results and offer some guidance on how their results should be read in light of those differences.

Author response:  The major difference between experiments 1a and 3 are that in experiment 1a, glyoxal was added in a pulse, and created a small, short-lived optical signal in the CAPS instrument, which is well-suited to measuring sudden changes in optical parameters.  In experiment 3, glyoxal was added continuously over 25 minutes to reach the same concentration, and was therefore not detectable by CAPS.  In both experiments, no increase in particle diameter was detectable by SMPS.  However, experiment 3 also had AMS monitoring, which recorded a barely-measurable 0.9 ug/m3 increase in organic aerosol (S/N = 1.6).  The manuscript has been edited to clarify these differences:

> "To better understand the reactive processes happening in the dry aerosol particles, further experiments were conducted in a 300 L Tedlar chamber probed by Q-AMS, SMPS, CAPS-ssa, and CRD/PAS.  Figure S2 shows an AMS ion correlation plot comparing signals before and after 40 ppb glyoxal was added over a period of 25 minutes to the dry chamber containing AS aerosol in experiment 3.  Unsurprisingly, the gradual addition of this smaller amount of glyoxal did not cause observable net particle growth or a decline in aerosol albedo at 450 nm.  A marginal (0.9 $\mu g/m^3$, S/N = 1.6) increase in total organic aerosol was observed by Q-AMS during glyoxal addition, associated with increases in ion signals at $m/z$ 15 ($CH_3^+$ or $NH^+$ fragments), 23 ($Na^+$), 29 ($CHO^+$ fragment), 47 (formic acid+$H^+$ or a $CH_3O_{2+}$ fragment), 69 (imidazole-$H^+$), 81 (pyrazine-$H^+$), 97 and 119 (imidazole carboxyaldehyde "IC" +$H^+$ and $Na^+$, respectively), while slight decreases in aerosol water signals were observed at $m/z$ 16"

2. In lines 95-99 the authors make the case that the lack of particle growth is consistent with a lack of uptake of glyoxal at low RH, but in lines 195-196, the possible (though small) uptake of glyoxal is highlighted as a reason for water depletion. If the results suggest that glyoxal is able to access surface water (and thus uptake to the particle even under dry conditions), shouldn't this be taken into account in the particle growth discussion?

We have edited line 95-96 to remove the comment about uptake, since what we were trying to point out is that the lack of observed growth under dry conditions was consistent with earlier studies.  In this work, however, our chemical and optical measurements suggest that the lack of growth does NOT indicate a lack of glyoxal uptake or reactivity.  Instead, they suggest that glyoxal is converting a fraction of the AS particles into brown carbon reaction products without a net gain in volume.  We now try to make this distinction clear from the outset of the results section.  This section now reads:

> "This lack of observed growth at <5% RH is consistent with previous studies under very dry conditions (Kroll et al., 2005;De Haan et al., 2017).  However, as optical and chemical measurements described below will show, this lack of growth does not indicate a lack of glyoxal reactivity."

3. Following on to point #2, in lines 102-104 the discussion is hard to follow. The loss of glyoxal is largely to the steel walls, and yet the large albedo decline is also due to glyoxal-driven surface reactions? Does this suggest that even minor amounts of reactivity lead to very high albedo changes? I think the confusion readers will have with this section that the authors need to clear

up is related to the sizing language. How should we read "largely", "large", and "at least some" in order to understand the weight of the argument the authors are making here?

We have re-written this section to try to avoid ambiguity and confusion. The rapid drop in albedo indicates rapid particle-phase brown carbon formation, which is followed by a much slower loss of glyoxal to the chamber walls. This paragraph now discusses these two events in order:

> "The addition of 0.05 ppm glyoxal gas at $t$ = 4:47 h triggered a short-lived drop in albedo by 0.034 that was observed by CAPS-ssa at 450 nm. A second, larger glyoxal addition (0.50 ppm) at $t$ = 5:16 h caused albedo to plummet to 0.75, a change 7× greater than the first. Even though particle sizes did not increase after either glyoxal addition, the significant albedo declines indicate that glyoxal reactions rapidly produced light-absorbing products at dry AS particle surfaces. Over the next 30 minutes, as glyoxal gas-phase concentrations decreased by half (likely to due to chamber wall losses), aerosol albedo recovered proportionately, indicating that this surface brown carbon formation under dry conditions is fully reversible."

4. In regards to Figure 1, how much of the recovery of albedo upon introduction of water is due to exclusively to the introduction of the water and how much is due to there still being glyoxal in the chamber as evidenced by the PTR-MS signals at 59 and 31? Unlike the earlier addition of a small glyoxal concentration, this larger addition was not allowed to return to baseline, and while this may not matter in the resultant albedo recovery upon water addition discussion, this overlap isn't addressed adequately in the text. Essentially, is the albedo increase due to the addition of the water, or the resultant dilution/loss of glyoxal signal?

Since loss of glyoxal caused a loss of brown carbon for the previous 30 minutes without water addition, this mechanism is presumably still in play when water addition causes most of the remaining glyoxal to be removed from the gas phase (likely due to rapid wall losses). However, we cannot rule out an additional contribution from water, due either to dilution effects (suggested by Reviewer 1) or hydrolysis of brown carbon products. We have added a new paragraph that discusses the mechanisms of brown carbon formation and removal upon humidification:

> "Proposed chemical mechanisms for brown carbon production at AS particle surfaces are summarized in Schemes 1 and S1. Except for steps where new N-heterocyclic rings are formed, all processes are reversible (Kampf et al., 2012). Thus, a reduction of gas-phase glyoxal concentrations will shift reversible reactions away from BrC back towards simple N-heterocycle products, which do not absorb 450 nm light. Humidification to 50% RH accelerates this shift by removing more glyoxal from the gas phase, and perhaps also by hydrolysis of double bonds and dilution effects (Rincón et al., 2010;Phillips and Smith, 2014, 2015)."

5. In lines 143-144, and Figure 2 the case is made for albedo change as a function of glyoxal concentration being a second order polynomial with respect to glyoxal concentration. Is there a physical explanation that would help defend this choice, or is it simply what fit the data? Or, is there a reason that using a two-slope approach (say a linear fit to the data at [glyoxal] < 0.35 ppm

and another linear fit to the data at [glyoxal] > 0.35 ppm) wouldn't also successfully capture the data? The authors should offer with what significance the readers should approach this polynomial fit and [gly]^2 dependence.

Since imidazole formation requires two glyoxal molecules, it is reasonable that the production of imidazole-based brown carbon is second order with respect to glyoxal concentrations. Furthermore, the kinetics of glyoxal + NH4+ are known to be concentration dependent, as determined by Noziere *et al.* (Noziere et al., 2009) We have added the following sentence justifying the curvature in Figure 2 to the manuscript:

> "While glyoxal + ammonium reactions are $1^{st}$ order in glyoxal in dilute solution when [glyoxal] $\times$ [NH$_4^+$] < 1.2 M, they switch to $2^{nd}$ order at higher concentrations (Noziere et al., 2009), which are likely in adsorbed surface water environments in these dry experiments."

Minor and technical comments:
1. At least when I downloaded a copy of the paper, Table 1 had some jumbled values in the last column (it would appear that those values are line numbers). The authors should check this Table to ensure that everything is in the place they expect it to be. This could very well be an artifact of the download and not the paper, and so this might disappear in final publication.

This appears to have been a function of the pdf conversion. We will pay attention to this.

2. In regards to the discussion on lines 195-196 again, a more curiosity-driven question the authors may consider, if they wish, commenting on is: can this ability of glyoxal to access surface water lead to a localized area of highly concentrated glyoxal (and that would thus accelerate chemical reactivity)?

Yes, in our opinion the evidence points in this direction. As described earlier, the kinetics of this reaction are concentration dependent, and in this study the kinetics clearly match high concentration conditions. We have added this idea to the discussion of Figure 2:

> "While glyoxal + ammonium reactions are $1^{st}$ order in glyoxal in dilute solution when [glyoxal] $\times$ [NH$_4^+$] < 1.2 M, they switch to $2^{nd}$ order at higher concentrations,(Noziere et al., 2009) which are likely in adsorbed surface water environments in these dry experiments."

We explain this concept in more detail at the end of the Results section:

[revised manuscript text omitted]